# Exploring Opportunities and Challenges of AI in Primary Healthcare: A Qualitative Study with Family Doctors in Lithuania

**DOI:** 10.3390/healthcare13121429

**Published:** 2025-06-14

**Authors:** Kotryna Ratkevičiūtė, Vygintas Aliukonis

**Affiliations:** Centre for Health Ethics, Law and History, Institute of Health Sciences, Faculty of Medicine, Vilnius University, 10257 Vilnius, Lithuania; kotryna.ratkeviciute@mf.stud.vu.lt

**Keywords:** family medicine, primary care, artificial intelligence, ethics, black box

## Abstract

Background and Objectives: AI is transforming healthcare, with family doctors at the forefront. As primary care providers, they play a key role in integrating AI into patient care. Despite AI’s potential, concerns about trust, data privacy, and physician autonomy persist. Little research exists on family doctors’ perspectives. This study investigates the views of Lithuanian family physicians on AI’s ethical challenges and benefits, aiming to support responsible implementation. Materials and Methods: A review of the literature was conducted (2015–2025) using Google Scholar, PubMed, and Scopus. This qualitative study explored family physicians’ perceptions of AI in Lithuania, focusing on ethics, AI’s role, experience, training, and concerns about replacement. Informed consent and ethical guidelines were followed. Results: AI has strong potential in family medicine, automating administrative tasks, improving diagnostic accuracy, and supporting patient autonomy. AI tools, like clinical documentation systems and smart devices save time, allowing physicians to focus on patient care. They also improve diagnostic precision, enabling earlier detection of conditions such as cancer and coronary artery disease. Physicians express concerns about AI’s reliability, biases, and data privacy. While AI boosts efficiency, many emphasize the importance of human oversight in decision-making, especially in complex cases. Privacy concerns around health data and the need for stricter regulations are crucial. Lithuanian family physicians generally accept AI as a helpful tool for routine tasks but remain cautious regarding its trustworthiness. Job displacement concerns were not prevalent, with AI seen as a tool to augment rather than replace their role. Successful AI integration requires training, transparency, and ethical guidelines to build trust and ensure patient safety. Conclusions: AI enhances efficiency in family medicine but requires structured training and ethical safeguards to address concerns about data privacy, accountability, and bias. AI is viewed as supportive, not as a replacement.

## 1. Introduction

“Can machines think?” This was the central question posed by Alan Turing in his 1950’s paper, “Computing Machinery and Intelligence”. This work marked the beginning of artificial intelligence’s evolution and the first significant phase of its development, often referred to as the “AI spring”. Turing introduced a concept now commonly known as “The Turing Test”. Through this experiment, he aimed to evaluate machine intelligence. The key idea was that if a machine could engage in a conversation where a person could not distinguish it from a conversation with another human, the machine would be considered intelligent. After widespread media attention, significant scientific breakthroughs in AI, and substantial investments, an “AI winter” set in—a period marked by disappointment and unfulfilled promises. This cycle of “winter” (recession) and “spring” (growth) repeated once more until the 1990s, a decade that saw progress in machine learning. These advancements eventually paved the way for deep learning, leading to the third AI spring, which began in 2010 and continues on today [1].

The healthcare system is constantly evolving, adapting to the challenges and changes the world faces. Changes are also driven by the rapid growth of medical data, which increases by approximately 48% annually [2]. Healthcare systems are continuously expanding their use of data and emerging technologies to enhance their capabilities. This transformation is often described as the rise in the “data-driven physician,” reflecting the increasing digitalization of health data and the growing role of AI in medical practice. In the future, clinicians are expected not only to use AI tools but also to actively contribute to their development and implementation [3]. As we stand at the start of these major changes, it is crucial to understand AI’s capabilities in healthcare, assess the current state of AI use, and begin educating doctors and healthcare professionals about the inevitable changes ahead.

Family doctors handle an immense amount of information, making AI adaptation training especially valuable [4]. Only family doctors embody all four Cs of primary care—first contact, comprehensive, coordinated, and continuous—described by Dr. Barbara Starfield, which are foundational to modern healthcare, and AI technologies are being integrated into every stage of these pillars. Family doctors are the first point of contact in the healthcare system, playing a vital role in early diagnostics; by applying AI in this step, decision-making can be enhanced and patient flows simplified. Comprehensive care reflects the broad spectrum of services family doctors provide, from routine checkups to complex interventions, where AI can assist in managing diverse clinical data and identifying care gaps and patient needs. As the central hub of healthcare, family doctors lead coordinated care, where AI can improve data sharing, communication, and overall management. Finally, continuous care highlights the long-term relationships family doctors build with patients, and AI can strengthen these bonds by offering personalized insights, fostering trust, and promoting ongoing engagement. These pillars collectively enhance the integration of AI, creating a more efficient and patient-centered healthcare system [5].

With the excitement and joy that new technologies bring also come significant challenges and ethical questions. Addressing these challenges is complex and depends on whose interests are represented. In 2019, a study conducted in China examined the challenges of using AI in healthcare. The participants—specialists from three key sectors: government policymakers, hospital managers, and IT firm managers—highlighted two primary ethical issues: trust and data sharing. Hospital managers and government policymakers noted that patients often lack trust in AI decisions, as they prefer direct communication with doctors over relying on AI-generated recommendations. Additionally, AI relies on vast amounts of data, which raises concerns about trust due to the absence of clear ethical guidelines. There is also concern that foreign companies might compromise national security by accessing sensitive patient data. The lack of regulations creates a chaotic market where AI algorithms often function as “black boxes”, making them difficult for users to understand. In contrast, IT firm managers downplay these concerns and instead focus on the importance of establishing performance standards to build trust in AI systems [6]. Ethical challenges are also tied to doctors’ fears about AI. Medical professionals worry about losing their autonomy and the ability to make important decisions [7]. They are concerned that their normal work routines will change and that their workloads will increase [8].

While there is a growing body of research addressing the integration of AI into clinical practice, relatively few studies have specifically explored the perspectives of family doctors, particularly regarding the ethical implications of AI in primary care. Given that family doctors play a crucial role within healthcare systems, often serving as the first point of patient contact and shaping broader public health practices, understanding their viewpoints is especially important. This study aims to contribute to filling this gap by investigating Lithuanian family doctors’ experiences, concerns, and expectations surrounding AI implementation. Insights gained from this research can help inform strategies for responsible AI adoption in primary healthcare, prioritizing patient safety, and improving healthcare quality.

## 2. Materials and Methods

A review of the literature was conducted to gain a broader perspective on the topic. A comprehensive search was performed across multiple databases, including Google Scholar, PubMed, and Scopus for articles published between 2015 and 2025 using predefined keywords and their combinations, such as “artificial intelligence,” “ethics”, “ethical challenges”, “EU regulations”, “medical AI”, “healthcare”, “medical imaging”, “machine learning”, “deep learning”, “black box”, and “family medicine”. Additionally, backward citation searching was utilized to ensure the inclusion of relevant studies. Then, a qualitative analysis of Lithuanian family medicine physicians was conducted to systematically code and explore family physicians’ perceptions of artificial intelligence (AI) in clinical practice. Using an inductive coding approach, we did not begin with a set hypothesis; instead, we let the data speak for itself and allowed recurring themes to emerge naturally from the interview transcripts. The literature review provided a broad understanding of both key application advantages and ethical, regulatory, and clinical concerns related to AI in healthcare. It also informed the overall direction of the study by highlighting topics that are currently debated in the field and likely to be relevant to clinicians. The qualitative phase allowed us to examine how these global topics are perceived, reinterpreted, or questioned by family physicians working in the Lithuanian primary care context, and whether they are considered meaningful in daily clinical practice.

All interviews were conducted remotely via Microsoft Teams to accommodate participants’ schedules and avoid logistical challenges. Before each interview participants were informed of the study’s objectives, the voluntary nature of participation, the confidentiality of their responses, and their right to withdraw at any time without consequence. The interview transcripts were edited to preserve the integrity of the language and capture the physicians’ perspectives on AI integration.

We used purposive sampling to recruit experienced family physicians from diverse institutional settings and with varying years of practice. This selection aimed to ensure a range of perspectives relevant to AI integration in Lithuanian primary care.

Qualitative data from the interviews were analyzed using thematic analysis following the framework of Braun and Clarke (2006) [9]. The process included familiarization with the data, generating initial codes, searching for themes, reviewing themes, defining and naming themes, and producing the report.

Data saturation was monitored continuously during the interview and coding process. After each round of interviews, both authors independently reviewed transcripts to assess whether new codes or themes were emerging. After the 14th interview, when both reviewers agreed that no new major themes had been identified, we conducted two additional interviews to confirm thematic redundancy. As these interviews did not yield novel findings, we considered saturation to be achieved.

Some codes were merged when overlapping themes were identified. This process allowed us to include the most meaningful and representative themes in the final analysis.

Ethical issues: concerns about privacy, responsibility, autonomy, doctor–patient relationships, trust, and bias.

Present and future of AI: defined as skeptical, neutral, or hopeful about AI’s role in clinical practice.

Personal experience: responses about using AI for personal tasks or work-related tasks (whether used or not).

Training: whether participants had training (had or did not have training) and whether they felt the training was needed or not needed.

Positive opinions: perspectives on AI’s benefits, such as improving patient outcomes, benefit for administrative work, and benefit for clinical work.

Negative opinions: responses about limitations and lack of trust in AI-generated results.

Fear of replacement: addressed whether physicians feared being replaced by AI (yes/no).

## 3. Literature Review

### 3.1. Technical Overview

Artificial intelligence, as historically defined, refers to machines designed to replicate human intelligence and, in some cases, surpass it by performing assigned tasks [10]. It serves as a broad framework that includes various interconnected technologies, such as machine learning and deep learning, which are used to develop tools like natural language processing (Figure 1).

Machine learning (ML) primarily focuses on developing algorithms that allow computer systems to learn from experience. These algorithms utilize training data to build models capable of making predictions or decisions without explicit programming.

ML is categorized into four main types: supervised, unsupervised, semi-supervised, and reinforcement learning. Supervised learning refers to algorithms trained on labeled data, where each input is linked to a corresponding output. This approach is widely used in healthcare. For example, a dataset might include images of skin lesions as input data and their corresponding diagnoses, such as “benign” or “malignant,” as output labels [11]. Unsupervised learning involves working with unlabeled data, where the algorithm explores the dataset to identify underlying patterns or structures. One common technique is clustering, which involves grouping similar elements. In healthcare, for example, clustering can be used to categorize patients into groups based on the similarity of their symptoms. Semi-supervised learning combines both labeled and unlabeled data. Reinforcement learning, on the other hand, involves algorithms that learn to make decisions by interacting with their environment and maximizing rewards based on feedback. For example, a system might learn to play a game by continuously playing against itself and improving its strategy [1]. In healthcare, this approach could be applied to robotic surgical systems, allowing them to refine precise movements, such as suturing wounds during surgery [11].

Deep learning (DL) is a subfield of machine learning that uses multilayered neural networks designed to mimic certain aspects of human brain function. These networks consist of artificial neurons that process data through adjustable weights and biases. The structure includes an input layer that encodes raw data, multiple hidden layers that extract hierarchical features, and an output layer that produces the final prediction or classification. The network learns through backpropagation, an iterative process that adjusts weights to minimize errors, improving accuracy over time [12].

DL algorithms power various types of neural networks, each designed to solve specific problems. Some specialize in image analysis, detecting features and patterns within images, and are applied in fields like computer vision. For example, they can be used to detect malignant tumors in patient X-ray images, providing secondary assessments and highlighting areas of concern [11]. Others generate synthetic images and data to support research and training. Certain networks process language and sequence data, making them useful for natural language processing (NLP) [12]. NLP plays a crucial role in structuring and analyzing medical text data, including electronic health records, diagnostic reports, and patient feedback. By breaking down and standardizing text, NLP enables tasks like extracting key medical information, summarizing patient histories, and supporting clinical decision-making [13]. These AI-driven technologies are increasingly integrated into smart healthcare systems, streamlining documentation, enhancing diagnostics, and improving patient care.

One of the major drawbacks of AI is its complexity. Most AI systems function like “black boxes”, making it difficult to understand their inner workings, data interpretation, or the reasoning behind their decisions. While this approach often delivers high accuracy, the lack of transparency creates challenges in establishing trust, interpreting results, and handling errors [14]. Since reliability is crucial for applying AI in medicine, it is essential to ensure that every stage of model development is clear and transparent [15]. In a 2020 paper, Loyola-Gonzales observed that while “black box” AI models can be difficult to understand, their results can still hold significant value if presented in a way that is clearly interpretable by specialists. For instance, healthcare professionals may not need to fully comprehend the inner workings of an AI system, but it is crucial that they receive conclusions in a format that is understandable and actionable. However, machine learning experts must have a thorough understanding of these models to optimize their performance and ensure their accuracy [16]. Explainable AI (XAI) aims to solve these challenges. According to Parvathaneni Naga Srinivasu et al., XAI relies on five main principles: explainability, which ensures transparent communication between humans and AI systems; understanding, which involves analyzing model components and behavior; fidelity, which evaluates how precisely the model explains its decisions; transparency, which ensures that a model’s decisions can be easily interpreted; and finally, adaptability, which refers to the model’s ability to quickly learn and adjust to new information or environments [15]. XAI’s potential is gaining recognition, especially among those focused on sustainability, transparency, and ethical AI. The SustAInLivWork project in Lithuania is one example, aiming to develop explainable AI (XAI) to make AI decisions more transparent, helping people understand the reasoning behind results and ensuring trust in AI-driven outcomes [17]. However, despite the potential of XAI, Bologna and Hayashi (2017) emphasize the need for a compromise between accuracy and explainability [18]. “Black box” methods, such as deep learning, are highly accurate but lack transparency, whereas XAI models are more explainable but less accurate. This balance is especially critical in medicine, where both high accuracy and high explainability are essential to build trust in AI systems among doctors [18]. However, some scientists, such as McCoy et al. in their 2021 paper, raise the question of whether it is always crucial to prioritize explainability. They argue that focusing too heavily on understanding AI systems could limit their potential to solve complex problems that surpass human capabilities [19]. As an alternative, Loyola-González suggests combining “white box” and “black box” methods to leverage their respective strengths. One proposed approach is the layered technique, in which the outputs of both methods are used as inputs to separate layers, with the results integrated into a single, unified output. Another approach is mutual result complementarity, where one method provides the core information, while the other method contributes additional insights to refine and optimize the final result [16].

### 3.2. EU Regulations

#### 3.2.1. The European General Data Protection Regulation (GDPR)

The General Data Protection Regulation (GDPR) safeguards individuals’ privacy by regulating the processing of their data. It grants them the right to easily control their personal information, transfer it between service providers, request deletion (the “right to be forgotten”), and be notified of data breaches. The regulation establishes uniform rules across the EU, reduces administrative burdens for businesses, and mandates the appointment of data protection officers when processing large volumes of sensitive data. It also requires the implementation of integrated data protection measures, including the use of pseudonymization and encryption [20].

Although the GDPR does not explicitly mention AI, its rules are highly relevant to AI, particularly in relation to the limitation of data processing purposes, protection of sensitive data, data minimization, and restrictions on automated decision-making. These rules may conflict with the need for AI technologies to process large volumes of personal data, the purpose of which may evolve from its original intent. While the GDPR seeks to align data protection with the development of AI, the absence of clear guidelines and the complexity of AI make this challenging. Effective implementation of the GDPR relies on clear guidance from data protection authorities to help companies make informed decisions that comply with the regulation’s requirements [21].

#### 3.2.2. The EU Medical Devices Regulation (MDR)

The EU Medical Devices Regulation (MDR) establishes rules for the marketing and use of medical devices to ensure a high level of patient safety and health protection. Devices are classified into four risk classes, from the lowest risk (class I) to the highest risk (class III). Medical devices include a wide range of products, including hardware, software, and implants, designed to diagnose, monitor, treat, or predict disease [22]. According to classification rules, most AI medical devices will be classified as Class IIa, unless they involve monitoring vital physiological parameters (e.g., respiration, blood pressure), in which case they will be classified as Class IIb. High-risk AI, where incorrect outputs could lead to serious harm or death, is classified as Class III. These devices require a notified conformity assessment body to ensure they meet the manufacturer’s declared performance and safety standards [23].

#### 3.2.3. Artificial Intelligence Act (EU AI Act)

The EU AI Act, which came into force in 2024, is the European Union’s most significant regulation on artificial intelligence. It classifies AI systems by risk level and sets specific requirements for their use. AI applications deemed to pose an unacceptable risk, such as social scoring or manipulative techniques, are strictly prohibited. High-risk AI, including systems used in medical diagnostics or treatment eligibility assessments, must undergo rigorous third-party evaluations, adhere to strict data management standards, and implement risk mitigation measures. Limited-risk AI, such as chatbots and deepfakes, must meet transparency requirements by informing users of their AI-driven nature. Minimal-risk AI, like video games and spam filters, remains unregulated, though developers may voluntarily follow the Act’s guidelines. The Act also introduces new obligations for general-purpose AI models (GPAI), such as GPT-4 and DALL·E. Developers of these models must provide technical documentation, disclose training data sources, comply with EU copyright laws, and ensure transparency. High-impact GPAI systems, which have broad societal influence, face additional requirements, including continuous risk monitoring and incident reporting [24,25].

Van Kolschooten and van Oirshot explore the implications of the Act for the healthcare sector. AI-powered medical devices, particularly those used for diagnosis, treatment, or emergency care, must comply not only with the AI Act but also with the existing EU Medical Device Regulation (MDR). The Act introduces additional obligations, including risk management, human oversight, and data governance, aimed at strengthening patient safety. While low-risk AI applications, such as wellness apps or mood-tracking tools, are subject to minimal transparency requirements, administrative hospital systems are not regulated. The Act also applies to healthcare organizations using AI, not just AI developers. For instance, hospitals that implemented AI-powered triage systems during COVID-19 must now ensure compliance based on the system’s risk classification [26].

### 3.3. AI Potential in Family Medicine

AI has the potential to improve many aspects of family medicine. Although research specifically focused on primary care is still emerging, it offers valuable insights that we have integrated into our review. At the same time, due to the novelty of the field and the limited number of in-depth studies, we also refer to findings from related areas of medicine and healthcare technology. Insights from fields such as clinical documentation, diagnostics, workflow optimization, and patient-centered care provide a foundation for understanding how AI could transform family medicine.

AI has immense potential to enhance patient-centered care and personalized medicine, the gold standard for clinical decision-making. By automating time-consuming tasks, improving diagnostic accuracy as a second look, and promoting patient autonomy, AI strengthens the doctor–patient relationship [10,27].

One of AI’s most impactful applications is assisting with clinical documentation. AI-powered systems can capture physician–patient conversations and generate structured notes, allowing doctors to focus more on direct patient care instead of administrative tasks [28]. Additionally, AI optimizes workflows, such as appointment scheduling, patient triage, and specialist referrals, reducing wait times and ensuring timely and appropriate care [29].

AI also enhances diagnostic accuracy, enabling earlier and more precise disease detection. By processing vast amounts of data, including years of clinical records, AI helps doctors make more informed decisions [11,29]. In some cases, AI has achieved diagnostic accuracy rates of 90–100%, matching or surpassing human doctors while reducing errors [30]. For instance, AI has outperformed traditional mammogram analysis in breast cancer risk assessment, improving the timing of follow-up screenings [31]. Similarly, deep learning models can automatically detect coronary artery plaques and analyze heart images, facilitating earlier and more accurate treatment [32]. While AI remains in the early stages of medical integration [33], its role as a “second look” for physicians continues to grow.

Beyond diagnostics, AI-powered tools enhance patient autonomy by shifting healthcare toward a more collaborative model. Smart devices and apps enable patients to monitor their health at home, tracking vital signs, fall risks, and medication adherence in real-time. This not only empowers patients to take an active role in their care but also supports doctors in monitoring disease progression more effectively [10,33]. AI-driven technologies like ChatGPT further extend this innovation, assisting with patient education, medication reminders, and personalized care plans [34,35,36]. Additionally, care robotics, especially in elderly care, offer promising solutions for mobility assistance and medication management [37,38].

### 3.4. Ethical Challenges

#### 3.4.1. Inaccuracy and Potential Errors

Despite significant advancements in AI and the growing availability of medical data, integrating AI into clinical practice comes with risks due to occasional inaccuracies and potential errors. These errors can lead to serious consequences, including false negatives where diseases go undiagnosed, false positives that may result in unnecessary treatments, and the misprioritization of interventions in emergency settings. Such issues often stem from key challenges, including noisy input data (common in ultrasound diagnostics), data shifts (when real-world data differs from training data), and invalid contexts, such as artifacts or faulty signals [10]. A major factor contributing to AI’s lack of accuracy is its “black box” nature, which makes decision-making processes difficult to interpret or explain. When an AI system malfunctions, the consequences can be severe, as these models cannot account for every possible scenario. Additionally, because AI models are typically trained on retrospective data, they may struggle to adapt to real-world clinical situations, raising concerns, especially for autonomous systems like diagnostic AI chat applications, which may provide recommendations without regular updates or sufficient oversight [38]. Finally, the effectiveness of AI in healthcare depends not only on its technological robustness but also on how it is implemented. Insufficient training for healthcare professionals, limited involvement of medical experts in AI development, and unclear validation processes for commercial AI tools all increase the likelihood of errors [10]. To reduce the risk of errors, it is essential to update training programs for healthcare professionals, focusing on the use of medical AI in clinical practice, and to improve AI literacy among the public. For instance, the University of Helsinki offers free courses to help individuals better understand AI technologies and their benefits [39].

#### 3.4.2. Privacy and Security Issues

The use of AI in healthcare raises significant concerns about personal data safety, privacy, and identity protection. While long-term health monitoring enables the collection of valuable data that can improve treatment and help predict health risks, questions remain about how this data is stored and whether it is being used ethically or for non-medical purposes [40]. Private companies seeking profit by using health data pose a significant danger. Without sufficient regulations, patients may lose trust in the system and withhold important information, potentially interfering with accurate diagnoses and effective treatment [41].

Moreover, health data stored in the cloud or third-party servers is a prime target for cyberattacks. Medical records are particularly valuable for both research and commercial purposes, increasing the risk of unauthorized access and misuse [38]. Some argue that health data should be freely accessible for scientific advancement, provided that patients are informed and consent is obtained [42]. However, patients often do not understand what they are agreeing to, since long and complex consent forms typically favor manufacturers rather than users [27]. AI further complicates this issue by making data use unpredictable, reducing transparency and limiting patients’ ability to control or withdraw consent [43].

Anonymization and de-identification methods aim to protect patient privacy. De-identification removes personal details but allows re-identification through special access codes, while anonymization eliminates all links to identity, exempting data from GDPRs and bypassing consent requirements [38]. However, as Garbuzova (2021) warns, advanced AI and facial recognition can reconstruct identifiable features, making anonymized data vulnerable to re-identification [44]. Similarly, Bak et al. (2022) highlight how different countries take opposing approaches to health data governance [45]. Some enforce strict consent rules to protect privacy, giving individuals full control over their data, which protects privacy but limits large-scale AI research. Others permit the use of secondary data without re-consent, which boosts AI research but raises ethical concerns about whether individuals truly control their information. These disparities hinder international data sharing, making it difficult to train unbiased AI models or create global disease registries [45].

For AI to be used ethically in healthcare, clear regulations, strict privacy protections, and robust security standards must be established. Aligning technological progress with patient rights requires a balance between innovation and ethical responsibility. Additionally, simplifying consent processes and educating patients about data usage would foster trust and ensure AI serves public interests rather than corporate exploitation.

#### 3.4.3. Lack of Accountability, Transparency, and Responsibility

The integration of AI in healthcare raises important questions about responsibility. Could AI itself be held legally accountable? Transparency is crucial for building trust and involves both traceability (comprehensive documentation of an AI system’s lifecycle) and explainability (the ability to understand AI decisions) [10]. While AI can be a powerful tool in diagnosis and treatment, it lacks moral reasoning and free will [38]. If an AI system malfunctions, then determining liability becomes challenging. Should the responsibility fall on the doctor, who did not create the algorithm, or the developer, who has no direct involvement in patient care [46]? In a 2023 paper, Jie Zhang and Zong-ming Zhang argue that if a physician correctly follows AI-generated recommendations but the technology itself is flawed, accountability should lie with those involved in AI development, including those managing the algorithm design and data labeling. However, doctors still bear responsibility for diagnosing patients and making final clinical decisions. Medical institutions also play a critical role in ensuring proper AI training and implementation [38].

Therefore, close collaboration between AI developers and medical professionals is essential to ensure that AI tools are designed to meet clinical needs and do not function as autonomous decision-makers [41]. Ultimately, final decisions must remain in the hands of healthcare professionals to safeguard patient safety and maintain trust in medical technology.

#### 3.4.4. Bias and Discrimination

Integrating AI into healthcare presents not only technical challenges but also social ones, with system bias being a key concern. Global healthcare inequalities, based on factors such as gender, age, ethnicity, income, education, and geographic location, remain a significant global issue. Poorly implemented AI solutions can worsen these disparities, as one of the main contributors to bias is the biased data used to train AI models [10]. If discriminatory patterns exist in the training data, AI systems will replicate and amplify them rather than eliminate them.

A 2024 paper by Kolfschooten and Pilottin highlights this issue, discussing AI tools like MidJourney and ChatGPT-4, which often reinforce stereotypes. In healthcare, deep learning models may generate highly realistic content but reflect biased assumptions about gender, race, and professional roles. For instance, nurses are predominantly depicted as female, while leadership positions are typically portrayed as male. Similarly, certain medical conditions and professions are often linked to racial or ability-based stereotypes [47].

AI-enabled insurance systems can unintentionally perpetuate discrimination by using data like credit scores or criminal records as proxies for race, making bias harder to detect even when race itself is not explicitly considered [48]. Additionally, AI lacks the ability to make truly individualized decisions and may override patient autonomy, leading to paternalistic decision-making, especially in areas where social and psychological factors are critical [49].

To address these challenges, AI should not be developed as a one-size-fits-all solution but rather tailored to meet the needs of specific regions and populations. This approach is essential for minimizing bias and ensuring that AI-driven healthcare systems serve all individuals fairly, regardless of their identity or social status.

#### 3.4.5. Lack of Trust

A major social challenge in the adoption of AI is the lack of trust. The ‘black box’ nature of AI algorithms, where decision-making remains unclear, undermines doctors’ confidence in AI and limits their willingness to use these systems in practice. Explainable AI (XAI) techniques, which help clarify the logic behind algorithms, could help address this issue [8,15,17].

Trust is also shaped by AI’s impact on doctors’ professional identity. Physicians often resist changes that threaten their competence, status, or autonomy. A 2022 study by Ekaterina Jussupow et al. found that both novice and experienced clinicians perceive AI as a challenge to their professional identity. Novice doctors worry about personal recognition, while experienced doctors are more concerned about their abilities. These concerns can influence career choices, specialty preferences, and even how critically physicians assess AI-generated decisions [50].

Despite AI’s potential to enhance efficiency and personalize care, some fear it could weaken doctor–patient relationships. Specialists argue that time saved through AI might be used to see more patients rather than strengthen existing relationships, especially in healthcare systems that prioritize efficiency over quality [49]. This could result in superficial communication and reduced empathy, which are both essential for effective treatment. Additionally, the use of robots in care settings raises concerns about social isolation, emphasizing the need for human–robot collaboration to ensure dignified and ethical care in aging societies [14]. To balance the adoption of AI with patient-centered care, doctors should not only learn how to use AI effectively but also develop stronger soft skills. This would allow technology to serve as a tool that enhances patient care while also strengthening doctors’ abilities in human relationships [40].

Public attitudes toward AI in healthcare also reveal skepticism. While AI is expected to improve medical treatment, many still prefer human decision-makers. A 2023 US survey found that nearly half of the respondents trust doctors who do not use AI and believe their medical records should be accessed only by human physicians, not AI systems [51].

To rebuild trust, education and training in AI technology are essential, as people are less likely to trust what they do not understand. One proposed solution is an AI passport, which would standardize the documentation and traceability of medical AI tools. This passport would include details about the model’s development, clinical use, data sources, evaluation results, usage statistics, and maintenance procedures [10]. Promoting transparency through such initiatives could help bridge the trust gap and ensure that AI is seen as a valuable and accountable tool in healthcare.

## 4. Qualitative Analysis Results

To complement the literature review with real-world insights, we conducted a qualitative analysis of Lithuanian family physicians’ perceptions of AI in clinical practice. A total of 16 family physicians participated in the qualitative phase of the study. The majority were women (62%), with men comprising 38% of the sample. Most participants (56%) worked in the private healthcare sector, while 37% were employed in the public sector. All respondents were fully licensed family physicians who had completed their residency training. The largest portions of the sample had either 5–10 years (31%) or more than 20 years (38%) of professional experience. Workload distribution among participants was relatively balanced: 25% worked less than half-time, 25% full-time, 25% at 1.25 FTE, and 25% more than 1.25 FTE. All interviewees were based in the capital of Lithuania—Vilnius.

Participants were selected purposively to include experienced family physicians working in a range of healthcare institutions. While the selection was not based on personal connections, efforts were made to ensure diversity in institutional affiliation and experience level to capture a broad spectrum of perspectives relevant to the study’s aims. The higher representation of private sector physicians may be attributed to their greater availability and willingness to engage in the study.

During the interviews, no strict definition of artificial intelligence was imposed. Instead, participants were encouraged to describe and reflect on AI in healthcare based on their own understanding and experience. As a result, their responses covered a broad range of technologies, including chatbots, diagnostic support tools, administrative automation, and applications integrated into electronic health records (EHRs).

Given the study’s context and aims, this analysis provided several in-depth insights. The following section presents the key findings, organized to address the research questions and illustrate the experiences of practitioners.

In interviews, many physicians reported that AI could enhance their work efficiency. This observation aligns with the literature, which affirms substantial time savings through AI-driven automation. For instance, one physician noted that AI can accelerate decision-making and enhance data organization. Another respondent observed that automation could improve tasks such as prescription writing and diagnostic accuracy. Physicians’ willingness to use AI in the workplace in the future is similar in both the public and private sectors. The only difference is that private institutions are more inclined to implement internal AI medical systems, while public institutions remain more conservative. Several participants pointed out financial and infrastructural barriers as key reasons for this divide. In private healthcare settings, some clinics had already begun adopting commercial AI solutions, while many public institutions still lacked basic digital infrastructure. As one physician explained: “*When I enter patient data into the computer, I’m still doing what I used to write by hand. Even basic computerization is lacking, like having the patient’s height or temperature automatically recorded when they enter the room. That wouldn’t even be AI—just automation. And we’re not even there yet. There just isn’t enough funding. In private practice, some of these solutions are already appearing*.”. These differences highlight not only a technological gap, but also a growing imbalance in resources and readiness between the two sectors. This could lead to broader systemic consequences, such as increased physician preference for private institutions, and further limiting access to care in the public sector.

Several physicians highlighted the challenges of heavy patient workloads. For example, one respondent noted that she sees 60–70 patients per day, while another described working long shifts that extend beyond regular hours. These views imply that, in a high-demand environment, the delegation of routine and administrative tasks to AI could free up valuable time for more critical clinical decision-making and patient care. Some expressed hope for AI’s ability to automate routine administrative duties, allowing physicians to focus more on complex clinical decision-making and patient care. While physicians remain cautious about relying solely on AI for critical decisions, they collectively recognize its potential for enhancing various work-related processes.

### Implications for Ethical Integration

Despite these practical advantages, there were concerns regarding reliability, transparency, and trust. Physicians raised issues about AI’s decision-making accuracy, particularly in complex cases, and broader systemic concerns, such as potential biases in AI-generated recommendations. Some respondents highlighted that AI models trained on biased datasets could perpetuate existing disparities in healthcare, potentially disadvantaging underrepresented patient populations. Several physicians expressed caution, with one respondent noting, “*I always double-check AI recommendations because, despite its speed, I cannot fully trust it to capture the full complexity of a patient’s condition*.”. Others warned of the risk of over-reliance, as one physician explained, “*Relying too heavily on AI insights can lead to a common human tendency—laziness. If something tells me ‘This is the answer,’ then that’s the answer. The specialist might stop assessing the situation critically and instead fully trust AI, increasing the risk of errors*.”. Although the literature often highlights AI’s technical accuracy, our findings reveal that in everyday practice, clinicians remain wary and insist on maintaining human oversight. Adding to these concerns, some physicians noted that current AI systems are designed for the general public rather than specifically tailored to medical professionals. One respondent argued that if they knew exactly which databases the information was drawn from and if the AI were precisely adapted exclusively for clinicians, then trust would increase. This demonstrates a need for greater transparency regarding data sources to increase trust levels among clinicians for AI systems.

Several people interviewed expressed concerns about ethics and the need for guidelines: “*We need stronger ethical guidelines and better training to ensure that AI protects patient privacy and does not harm the doctor-patient relationship*.”. This opinion supports the call for better regulations, while highlighting the immediate need for ethical protections. However, some physicians did not share these concerns, reasoning that if AI has been approved for use by the European Union, it must already meet safety and regulatory standards, eliminating the need for additional worry. These differing perspectives highlight the importance of critical thinking in medical practice and the need for physicians to remain cautious rather than blindly trusting new technologies.

Interviewed physicians consistently viewed AI as a supportive tool that enhances their work without threatening their essential role in patient care. They emphasized that, while AI can efficiently automate routine tasks, it lacks human intuition and empathy—what one respondent described as the missing “fifth sense” irreplaceable in medical practice: “A*I can help with routine tasks, but it does not have that ‘fifth sense’ needed for true patient care*.”. This lack of fear relates to concerns about accountability. One interviewee noted, “*No matter how advanced AI becomes, the final responsibility is mine. Only a human can truly understand a patient’s context.*”. This sentiment reflects a strong belief in the importance of human oversight; despite AI’s significant role in decision support, the ultimate accountability for clinical decisions remains firmly with the physician. Consequently, contrary to broader predictions of workforce reductions driven by AI, our findings highlight that family physicians see AI as an augmentation of their expertise rather than a replacement. This perspective aligns with contemporary discussions in AI ethics that stress the value of maintaining human judgment and compassion in healthcare, ensuring that technology complements rather than displaces clinical practice.

While AI was largely seen as a supportive tool, some participants raised deeper concerns about its role in ethically complex situations. One respondent reflected on the ethical complexity of end-of-life decisions: “*For example, is it rational to treat an elderly person with a poor prognosis? (…) These questions arise in hospitals, wards, intensive care. And AI will say: resuscitating this patient is ineffective—better to stop. And someone will have to agree with these decisions.*”. Such reflections highlight physicians’ discomfort with AI potentially guiding care decisions in situations that require nuanced moral judgment. These concerns are not merely about accuracy, but about the risk of reducing complex human experiences to standardized algorithmic outputs. As a result, maintaining professional autonomy becomes crucial, not only to preserve clinical responsibility, but also to ensure that ethical and compassionate dimensions remain central in medical decision-making.

The theme of personal experience showed some interesting results. Forty-four percent of all participants use AI to some extent in their workplace, while only six percent do not use AI at all, either at work or in their personal lives. Several respondents discussed whether they used AI and how their personal use of digital tools influenced their clinical practice. For example, one participant noted, “*Using AI for personal tasks has made me more aware of its potential benefits at work, yet I still feel I lack adequate training to fully integrate it into patient care.*”, highlighting a gap between personal familiarity with the technology and its professional application. This insight points to an opportunity for improving clinicians’ education and support systems. Many physicians emphasized the need for specialized training that introduces official tools specifically designed for the medical community. Some respondents reported using AI at work for various tasks, primarily transcription, summarization, translation, and appreciated having AI as a second look. On the other hand, a particularly thought-provoking perspective came from a physician who expressed reluctance to share her personal clinical expertise with AI. She explained that integrating her hard-earned knowledge into an AI system would primarily benefit the institution, offering her little personal gain. This viewpoint highlights a key concern: if physicians are unwilling to transfer their unique insights to AI, even the potential for automating routine tasks may not fully materialize for individual practitioners, let alone the broader goal of developing more human-like systems. Some clinicians also questioned AI’s efficiency in clinical practice because response times are too slow. One respondent explained that she finds it more efficient to manually complete documents and forms than to wait for an AI to generate responses “*It takes me less time to write the status myself than to wait for the AI to generate a response, which takes 15–20 s*.”. This comment highlights two things. First, AI-generated output is perceived as inefficient compared to the speed of manual entry. Second, overload in a high-pressure clinical environment is so significant that the delay of 15–20 s seems like a serious issue for clinicians. These diverse perspectives suggest that while medical professionals are interested in AI, they remain skeptical of its reliability and efficiency, highlighting that the technology is not yet advanced enough to fully meet their clinical needs.

## 5. Discussion

This study contributes to growing international discussions on the ethical and practical integration of AI in healthcare. As emphasized in the WHO Global Strategy on Digital Health (2020–2025), national-level approaches are essential to ensure that digital technologies align with the needs of health workers and systems [52]. Our findings offer insights into how such global priorities are experienced and interpreted at the local level by family physicians.

When comparing our findings with the literature, substantial agreement exists on the benefits of increased efficiency and improved patient engagement. Respondents agreed that automating routine tasks can save valuable clinician time and enhance remote care. However, a split arises around reliability and trust. Whereas the literature often emphasizes AI’s technical precision [9,30,31,32], our data reveals that family physicians remain cautious, highlighting the need for continuous human oversight and practical training. Our respondents are concerned that while rules like GDPR set standards for protecting data, better safeguards and stronger encryption are needed to ensure data security in clinical settings.

Our findings align with several recent studies focused specifically on AI in family medicine. For instance, a 2023 empirical study conducted with family physicians in Saudi Arabia identified performance, expectancy, and trust as key factors in AI acceptance, and found that privacy concerns reduce willingness to adopt [4] patterns that also appeared in our data. However, unlike their study, which reported greater skepticism among older physicians, our respondents, including senior doctors, were generally supportive of AI even if some felt less confident about adopting it because of age-related comfort with technology. As one senior participant noted: “*AI has a bright future and could ease doctors’ work. Older people are more afraid of it, while young people keep up with technology anyway*.”.

These results also echo Steven Lin’s findings, who argues that primary care should lead the development of ethical, human-centered AI and promote implementation through appropriate training and regulations [5]. Our interviewees emphasized similar needs, particularly the importance of officially approved, trustworthy tools and structured education. One physician commented: “*It would be good if there were official or recommended AI tools, so I could trust it to really do the job.*”.

In line with the scoping review on AI in primary care by Kueper et al. [33], our participants acknowledged the relevance of AI in diagnostics and the use of EHRs. However, a key difference was that Lithuanian family physicians placed even more emphasis on the value of AI in handling administrative and routine tasks. This likely reflects the practical workload challenges faced in primary care, where the administrative burden is especially pronounced.

To our surprise, contrary to the concern in the literature, that being physicians’ fear being replaced by AI [50], none of our respondents expressed any anxiety about job displacement. Instead, they viewed AI as a tool to enhance their work rather than a threat. One physician stated, “*I am not worried that family doctors will disappear or that there will be fewer of us.*”. This observation indicates that, in our sample, physicians focus more on how AI can augment their clinical capabilities and routine tasks, rather than fearing a reduction in demand for their roles. Physicians also drew a clear line between what technology can offer and what only a human can provide, such as intuition, emotional support, and relational continuity. While the holistic approach is a defining strength of family medicine, it simultaneously presents unique challenges for the integration of artificial intelligence. The most interpretative and contextual aspects of clinical assessment—those centered on individual patient needs, preferences, and psychosocial background—are precisely the areas in which algorithmic standardization remains most limited. Algorithms excel at processing structured data and recognizing patterns but may struggle to accommodate the nuanced, relational, and dynamic factors that family doctors routinely consider in their patient-centered decision-making. This limitation raises important questions about the current and future scope of AI in family medicine, particularly regarding complex or ethically sensitive situations where individualized judgment is vital. Thus, rather than facilitating seamless integration, the holistic model may actually reveal the boundaries of what AI can (and cannot) support in everyday clinical practice [5,33]. Furthermore, although extensive ethical frameworks and regulatory measures are outlined in academic sources, our respondents stress the immediate need for clearer guidelines within the real-world context of family medicine.

As Kueper et al. [33] note, ‘Primary care settings across different countries vary substantially in their digital infrastructure, data availability, and integration of AI tools. This heterogeneity affects both the opportunities and barriers for implementing AI in primary care.’. Similarly, Lin [5] emphasizes that ‘the implementation of AI in primary care will depend on local factors, including existing digital health infrastructure, provider training, and regulatory environment.’. Our findings align with other researchers from Lithuania [53], who found that ‘private clinics [in Lithuania] were generally more open and better resourced to experiment with AI tools, while public institutions faced greater infrastructural and bureaucratic barriers.’. These results demonstrate the importance of considering local healthcare system features and digital maturity when assessing physician attitudes toward AI.

## 6. Limitations

This study is based on a small qualitative sample of family physicians in Lithuania, which limits the generalizability of the findings to other healthcare systems. While efforts were made to ensure variation in experience levels and institutional contexts, the sample may not reflect the full diversity of perspectives across the profession. Additionally, as interviews allowed participants to interpret the term “artificial intelligence” based on their own understanding, responses encompassed a broad range of technologies. While this openness enriched the data, it may have introduced variation in how participants framed their responses. Finally, as with all qualitative research, the findings reflect the context and time in which the data were collected and are best understood as exploratory insights rather than definitive conclusions.

## 7. Conclusions

In conclusion, our interviews reveal that AI in family medicine enhances operational efficiency and improves patient interactions by automating routine tasks, freeing clinicians to focus on complex decision-making and interpersonal care. However, successful integration of AI requires more than just technological deployment; it necessitates the development of structured training programs and the establishment of clear ethical safeguards to address data privacy, accountability, and bias risks. Although physicians appreciate the efficiency gains provided by AI, they remain cautious about its reliability and insist on maintaining ultimate clinical responsibility. They view AI as a complementary tool that supports, rather than replaces, human expertise.

This study connects theoretical ideas with practical experiences, providing valuable insights for future research.

## 8. Recommendations

While our study is based on a small qualitative sample, it provides timely insights into how family physicians perceive AI in clinical practice. Given the rapid pace of AI adoption in healthcare, early reflection on these perspectives is essential. The following practice-oriented considerations are not universal solutions, but grounded suggestions to guide further discussion, training, and policy development.

Prioritize Critical Thinking and Human-Centered Skills. While AI can automate technical tasks, physicians must remain actively engaged in decision-making and avoid blindly trusting AI-generated recommendations. Strengthening soft skills, such as communication and empathy, will emphasizing the irreplaceable role of human clinicians.Implement Structured AI Training tailored specifically for physicians. Healthcare institutions and businesses should develop formal educational programs to equip physicians with essential AI tools knowledge. Although many physicians express curiosity about AI, they often lack structured information and medical guidelines on AI tools. Training programs should bridge this gap.Understand AI Regulations and Risks. AI is not just a simple tool; it carries regulatory responsibilities. Physicians must be aware of existing AI regulations and the risks associated with its use, including potential data privacy breaches and ethical concerns.Develop a Basic Understanding of AI Technology. To use AI effectively and responsibly, physicians should familiarize themselves with its fundamental principles. Recognizing AI’s capabilities and limitations will enhance clinical decision-making and prevent unrealistic expectations.Take an Active Role in AI Development and Implementation. Physicians should contribute to AI development by proposing new functionalities for their institutions or engaging in AI-related discussions and projects. While programming skills are not necessary, proactive involvement can help shape AI applications that align with clinical needs.

## Figures and Tables

**Figure 1 healthcare-13-01429-f001:**
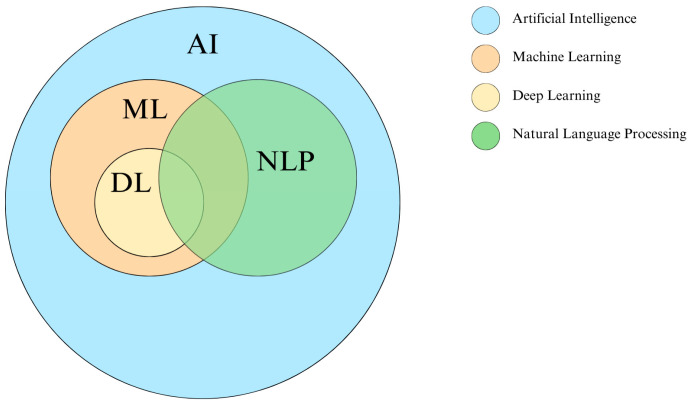
Interconnected AI technologies.

## Data Availability

Data contains in-depth interviews with clinicians. All participants agreed to data usage for article purposes, but the raw data supporting the conclusions of this article will be made available by the authors on request.

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
