# Peer review of "Exploring Opportunities and Challenges of AI in Primary Healthcare: A Qualitative Study with Family Doctors in Lithuania"

_healthcare, 2025, doi:10.3390/healthcare13121429_

Round 1
Reviewer 1 Report
Comments and Suggestions for Authors
I would like to see the discussion section expanded and include more material on the ethical implications of using AI in family care. For example, the authors should specify in some detail what kind of AI is being referred to when they interviewed the family doctors and they should report on some specifics of how AI is used by the doctors. However, these are minor and the authors are not required to submit a revised version.
Reviewer 2 Report
Comments and Suggestions for Authors
Dear Authors,
Thank you for submitting this timely and important manuscript. Your focus on AI integration in family medicine is both current and needed. The manuscript offers a well-balanced perspective, incorporating technical, ethical, and behavioral dimensions. I appreciate your efforts to explore physician perspectives based on real-world insights. Below are several suggestions that I believe will help strengthen the manuscript further:
- Clarify qualitative methodology
The qualitative component lacks detail. It is unclear how participants were sampled, how saturation was assessed, and which method was used for thematic analysis.
Suggestion: Add a subsection specifying:
- Participant number and selection method (e.g., purposive sampling)
- Thematic coding process (e.g., Braun & Clarke)
- Ethical approval reference (if applicable)
- Contextualize findings within global digital health frameworks
The manuscript does not reference key global initiatives like the WHO Global Strategy on Digital Health or the OECD AI Principles.
Suggestion: Briefly link your findings to at least one international policy framework to broaden the global applicability of your conclusions. - Address generalizability limitations
As all interviews were conducted with Lithuanian physicians, the findings may not generalize beyond that setting.
Suggestion: Acknowledge this limitation in the discussion and, if possible, compare briefly with findings from other regions. - Support narrative claims with evidence
Some statements—such as “physicians fear losing autonomy”—are presented without quotes or references.
Suggestion: Support such claims with either participant quotes or relevant citations. - Ensure consistent terminology
The manuscript inconsistently uses AI-related terms like "bias", "algorithmic bias", and "machine bias".
Suggestion: Consider adding a glossary or ensuring uniform use of terms throughout the text.
Minor Language and Structural Suggestions:
- Clearly state in the abstract that the interviews were conducted in Lithuania.
- Consider rephrasing some statements for clarity. For example:
❌ “Physicians may not fully trust AI.”
✅ “Physicians expressed conditional trust in AI, particularly regarding administrative tasks.” - You may also wish to group ethical reflections under a distinct subheading like “Implications for Ethical Integration” to increase visibility.
Final Recommendation: Minor Revisions
This manuscript provides an important and timely contribution to the intersection of primary care, medical ethics, and digital health. With minor revisions to improve methodological clarity, consistency, and international framing, the paper will be well-positioned for publication.
Comments on the Quality of English Language
Minor Language and Structural Suggestions:
- Clearly state in the abstract that the interviews were conducted in Lithuania.
- Consider rephrasing some statements for clarity. For example:
❌ “Physicians may not fully trust AI.”
✅ “Physicians expressed conditional trust in AI, particularly regarding administrative tasks.” - You may also wish to group ethical reflections under a distinct subheading like “Implications for Ethical Integration” to increase visibility.
Reviewer 3 Report
Comments and Suggestions for Authors
- Methodological description is not sufficiently clear. There is no explicit detail on participant selection, demographic diversity, or how the interviews were recorded and analyzed. The coding framework is not sufficiently detailed, this limit study replicability and produce potential bias in thematic extraction.
-The study is situated in Lithuania, but the manuscript implies broader implications for AI adoption in primary care. However, healthcare systems differ widely in digital readiness, infrastructure, and physician roles. Without comparative reflection, it is unclear to what extent the findings are generatable beyond the Lithuanian settings. A brief consideration of how local context may influence physician attitudes would improve international relevance.
Reviewer 4 Report
Comments and Suggestions for Authors
The manuscript presents a very current, timely, and pertinent proposal.
The content is presented and systematised in a consistent way, and the presentation is clear, both in terms of the objectives and the development of the argument.
Although there are no major weaknesses in this proposal, it is possible to highlight some critical points that should be improved or clarified.
Starting with the title, it is suggested that it could be rethought to more accurately reflect the scope of the proposal. Specifically, it would be important to make the methodological profile explicit (a qualitative study, although preceded by a literature review), as well as the limited scope of its development (Lithuania).
It would also be important to provide more information on the ethical appraisal and approval of this study.
Regarding what is stated between lines 69 and 70, the argument that the holistic approach of family doctors allows for a more effective application of AI is not persuasive enough. This point can be highly problematic because the most interpretative aspects of clinical assessment, centred on the patient and their context, are precisely those in which it is most difficult to standardise decisions based on information generated by algorithms. It's a sufficiently controversial point to be taken for granted.
Issues of professional autonomy are mentioned a few times (lines 12-13; 92; 432) but are very poorly substantiated and discussed based on empirical information.
Regarding point 2 - Materials and methods - there is not enough elaboration on the articulation between the two stages of the study, i.e., the literature review and the interviews. How did the first stage shape the next? There is also little information on the operational criteria for choosing the sample and recruiting participants. The characteristics of the interviewees are also unknown (men, women, age, years of professional experience, public or private sector). About this last differentiation, this is indicated (lines 471-472) but not discussed.
And how many participants are there anyway? There are 14, but it is also stated (line 122) that more interviews were carried out. How many more?
In lines 288 to 290 it is explained that examples from various areas of medicine have been used, given the novelty of the topic and the scarcity of research, however there are some bibliographical references (namely 4, 5, 27, 31 and 47) that would allow for a more in-depth discussion of the parallels and comparisons that can be made with this study. This level of discussion, which is referred to in line 556, should therefore be more in-depth, to highlight what this empirical evidence does or does not have in common or differentiates it from similar research or research centred on the same professional area.
Lines 513 to 520 revisit the issue of the holistic approach that is fundamental to the professional identity of family doctors. Here, too, this aspect could be further explored and discussed.
Lines 526 to 527 refer to a percentage, but this is not very enlightening because, as already mentioned, little is known about the sample.
Finally, on point 6, what is the point of putting forward recommendations? Their importance is not in question, but what is the basis for this option? Isn't the evidence from this study too limited to move in this direction?
Round 2
Reviewer 3 Report
Comments and Suggestions for Authors
I would like to thank the reviewer for adressing my comments